# Adaptive Neural Fault-Tolerant Control for the Yaw Control of UAV Helicopters with Input Saturation and Full-State Constraints

**Qiang Zhang [1,\*], Xia Chen [1] and Dezhi Xu [2]**

[1]  School of Electrical Engineering, University of Jinan, Jinan 250000, China; 13253528209@163.com
[2]  Institute of Electrical Engineering and Intelligent Equipment, School of Internet of Things Engineering, Jiangnan University, Wuxi 214122, China; lutxdz@126.com
\*  Correspondence: ave_zhangq@ujn.edu.cn

**Abstract:** In this paper, an adaptive neural fault-tolerant tracking control scheme is presented for the yaw control of an unmanned-aerial-vehicle helicopter. The scheme incorporates a non-affine nonlinear system that manages actuator faults, input saturation, full-state constraints, and external disturbances. Firstly, by using a Taylor series expansion technique, the non-affine nonlinear system is transformed into an affine-form expression to facilitate the desired control design. In comparison with previous techniques, the actuator efficiency is explicit. Then, a neural network is considered to approximate unknown nonlinear functions, and a time-varying barrier Lyapunov function is employed to prevent transgression of the full-state variables using a backstepping technique. Robust adaptive control laws are designed to handle parameter uncertainties and unknown bounded disturbances to cut down the number of learning parameters and simplify the computational burden. Moreover, an auxiliary system is constructed to guarantee the pitch angle of the UAV helicopter yaw control system to satisfy the input constraint. Uniform boundedness of all signals in a closed-loop system is ensured via Lyapunov theory; the tracking error converges to a small neighborhood near zero. Finally, when the numerical simulations are applied to a yaw control of helicopter, the adaptive neural controller demonstrates the effectiveness of the proposed technique.

**Keywords:** non-affine nonlinear system; adaptive neural control; actuator fault; full-state constraints; input saturation

## 1. Introduction

In the control field, with the rapidly expanding helicopter technology, the unmanned-aerial-vehicle (UAV) helicopter has been of wide concern in recent years; it has been applied to maritime supervision, environmental monitoring, search and rescue, agricultural and forestry protection, pipeline inspection, and aerial photography, to name just a few areas [1]. Since UAV helicopter flight control is a highly nonlinear, strongly coupled, and inherently unstable problem and subject to the uncertainties of various environments and varying flight conditions, the total dynamics of a UAV helicopter system are extremely complex, which have been decomposed into the longitudinal and lateral dynamics in [2–4], we only consider the turning movement of the UAV helicopter yaw control system under the comprehensive actions of actuator faults, input saturation, full-state constraints, and external disturbances in this paper. A linearized model cannot fulfill a global model approximation. Looking further, a nonlinear model for the yaw channel dynamics of helicopter is normally non-affine and has a control input that acts on the system in an implicit nonlinear way. As a result, it is a challenging task to determine the control input. To overcome this design difficulty for

a non-affine system, the traditional approaches contain an inverse control strategy [5] that requires more accurate mathematical models. T-S fuzzy control [6], mean value theorem [7–9], which has many online adjustment parameters, and Taylor series expansion [10]. In this paper, a combination of a Taylor series expansion with a robust sliding mode differentiator is employed to convert the non-affine nonlinear system into an affine nonlinear system.

Approximation-based adaptive control for multifarious nonlinear systems with unknown functions has drawn extensive research recently, and a great many schemes have been proposed. Fuzzy logic systems using linguistic information were applied to the control of unknown nonlinear systems in [11,12]. A direct adaptive fuzzy robust control method was described in [13] to cope with the problems caused by the dynamic uncertainties in single-input and single-output (SISO) strict-feedback nonlinear systems. On the basis of an adaptive fuzzy control and backstepping technique, a robust adaptive fuzzy backstepping stabilization control strategy was developed for a class of stochastic nonlinear switched systems in [14]. To approximate unknown nonlinear functions and improve control system robustness, a radial basis function (RBF) neural network (NN) was used to approximate nonlinear functions in an active power filters dynamic model [15]. An adaptive fuzzy neural network (FNN) control scheme based on an RBFNN was proposed [16] to enhance the robustness and compensation performance of the system. In [17], a fuzzy sliding mode controller based on an RBFNN controller was achieved for a three-link robot system. To take advantage of neural network online approximation performance, dynamic learning from neural control for a class of nonlinear strict-feedback systems with predefined tracking performance attributes was put forward [18] and then employed on a third-order one-link robot. In [19], output feedback adaptive NN controls were studied for two classes of nonlinear discrete-time SISO systems with unknown control directions. By combining backstepping and dynamic surface control with adaptive fuzzy state-feedback control, an adaptive fuzzy dynamic surface control was investigated for a class of nonlinear systems subject to a fuzzy dead zone, unmodeled dynamics, and unknown control gain functions in [20]. Nevertheless, these adaptive control schemes do not account for the combined function of input saturation and actuator fault.

As a result of space limitations, energy, and actuator physical performance, input saturation is ubiquitous in real-world control systems. Failures and faults are caused by actuators and sensors because of their continuous operation for long periods and unexpected external disturbances. Ignoring these factors, which can degrade nominal closed-loop performance, can cause a controller design to fail to achieve the desired tracking and even lead to instability. In [21], a second-order dynamic terminal sliding mode control was proposed for a class of non-affine nonlinear systems designed for input constraints and external disturbances. By adding a power integrator and backstepping technique, [22] devised a novel finite-time attitude control scheme for a rigid spacecraft subject to actuator saturation. In [23], the authors studied flexible-joint robot systems with input saturation and investigated an adaptive fuzzy dynamic surface control approach. Adaptive fault-tolerant control (FTC) has been used far and wide [24–26]. In [27], active adaptive fault-tolerant neural control was discussed for mitigating actuator fault problems in large-scale uncertain systems. By introducing a backstepping technique to fault-tolerant control, an adaptive actuator fault compensation control was studied in [28] for a class of uncertain multi-input single-out discrete-time systems with triangular forms. In [29], hybrid fuzzy adaptive FTC was presented for a class of uncertain nonlinear systems with unmeasured states. In [30], an adaptive neural-fuzzy sliding-mode fault-tolerant control was developed for uncertain nonlinear systems to handle actuator effectiveness faults and input saturation. Currently, there are still rare conclusions about non-affine nonlinear systems that can tolerate input saturation and actuator faults.

Moreover, the state constraints in the system are also an extremely significant matter. The Barrier Lyapunov function (BLF) is an effective tool to prevent the violation of constraints [31]. Adaptive control was proposed by designing a combined adaptive controller and BLF in [32–35] to satisfy the output constraints. In [36], an adaptive neural control was addressed for a class of stochastic pure-feedback nonlinear time-delay systems with unknown direction control gains

and full-state constraints. The work of [37] studied adaptive fuzzy tracking control-based barrier functions of uncertain nonlinear multi-input multi-output (MIMO) systems with full-state constraints; these systems have been applied to chemical processes. In [38], robust adaptive backstepping control for a class of non-affine nonlinear systems with full state constraints and input saturation was proposed. However, actuator faults have not been studied in the above literature.

To sum up, in this paper, we propose an adaptive neural fault-tolerant control scheme for a UAV helicopter yaw control system that provides for actuator faults, input saturation, full-state constraints, and external disturbances. The non-affine nonlinear system is converted to an affine nonlinear system via a Taylor series expansion and a robust sliding mode filter. The unknown nonlinear function can be approximated by an RBFNN, and furthermore, the scheme deals with bounded disturbances. Next, an anti-saturation compensator is used to analyze the impact of input constraints; full-state constraints issues can be managed by combining BLF with a backstepping design technique. Then, the Lyapunov theory is applied to verify that the proposed adaptive anti-saturation tracking controller can ensure the boundedness of all signals in the closed-loop system. The remainder of this paper is organized as follows: Section 2, the yaw dynamic of helicopter is given, a novel dynamic model transformation technique and the control scheme are proposed, and simulation results are presented to show the effectiveness of the proposed technique. Section 3 draws conclusions of this paper.

## 2. Main Results

In this section, Section 2.1 gives the modeling of UAV helicopter yaw-channel, the normal SISO non-affine nonlinear system structure is introduced in Section 2.2, the controller design and stability analysis of the closed-loop system are addressed in Section 2.3, illustrative examples are provided in Section 2.4.

### 2.1. UAV Yaw-Channel Model

In UAV helicopters, which are distinct from other types of robots because of their small-scale structure, the torque associated with the yaw control channel is provided with high sensitivity. To enhance the performance of helicopter yaw control, we consider a more precise model to characterize the yaw channel. In this paper, the model is adopted from [39], the framework of a rigid body UAV helicopter.

Based on [10], the yaw channel dynamic equation is described as

$$
\begin{cases}
\dot{\varphi} = r \\
I_{zz}\dot{r} = -Q_{mr} + T_{tr}l_{tr} + b_1 r + b_2 \varphi
\end{cases}
\tag{1}
$$

where $\varphi$ and $r$ are the helicopter's yaw angle and angular rate and $I_{zz}$ is the inertia around the z-axis. The z-axis is perpendicular to the design axis of the helicopter and points at the nose and below the fuselage in the helicopter's symmetrical plane. $Q_{mr}$ is the torque of main rotor, $T_{tr}$ is the thrust of tail rotor, $l_{tr}$ is the distance between the tail rotor and z-axis, and $b_1$ and $b_2$ are damping constants.

By using the blade element method [39], the torque $Q_{mr}$ can be formulated as

$$
Q_{mr} = \int_{R_0}^{R} \left( \frac{\rho \Omega^2 r^2 C_l c \phi}{2} + \frac{\rho \Omega^2 r^2 C_d c}{2} \right) r dr
\tag{2}
$$

therein, $\phi = v_1/(\Omega r)$, $C_l = a\alpha$, $C_d \approx C_{d0} + C_{d1}\alpha + C_{d2}\alpha^2$, where $\rho, a, r, \alpha, c, v_1, \phi$ and $\Omega$ are the density of air, slope of the lift curve, speed radial distance, angle of attack of the blade element, chord of the blade, induced speed, inflow angle, and rotor speed of the main rotor, respectively.

The non-affine nonlinear yaw-channel model is rewritten as

$$
\begin{cases}
\dot{\varphi} = r \\
I_{zz}\dot{r} = b_1 r + b_2 \varphi - (k_{Q_2}\theta_{mr}^2 + k_{Q_1}\theta_{mr} + k_{Q_0}) + (k_{T_2}\theta_{tr}^2 + k_{T_1}\theta_{tr} + k_{T_0})l_{tr}
\end{cases}
\tag{3}
$$

where $k_{Q_2}, k_{Q_1}$ and $k_{Q_0}$ are decided by the shape of the blades and the speed of the main rotor, $\theta_{mr}$ is pitch angle of main rotor, $k_{T_2}, k_{T_1}$ and $k_{T_0}$ depend on the shape of the blades and the speed of the tail rotor, $\theta_{tr}$ is pitch angle. According to Equation (3), the yaw dynamics of a UAV helicopter can be represented by a second-order time-varying non-affine system with input nonlinearity. The input nonlinearity is mainly caused by the main rotor collective, the speed of main rotor, and the speed of tail rotor.

The deduction process of non-affine nonlinear helicopter yaw-channel model is given in the Appendix A.

## 2.2. Normal Model

In general, we consider the following normal SISO non-affine nonlinear system structure:

$$\begin{cases} \dot{x}_1 = x_2 \\ \dot{x}_2 = F(t, x, u_{fs}) + D \end{cases} \tag{4}$$

where $x = [x_1, x_2]^T \in \mathbb{R}^2$ is measurable system state, $F(t, x, u_{fs})$ denotes a known smooth nonlinear function, $D$ is external disturbance. $u_{fs}$ is faulted saturation control input which is $u_{fs} = \rho u_s$, and $\rho$ is an unknown function that satisfies $0 < \rho \le 1$, $u_s$ is a saturation input, which is shown in the Appendix B $u$ denotes an actual control input to the system. On account of the limited workspace and security considerations, the states of the system are constrained and need to satisfy: $|x_1| \le \alpha_1$, $|x_2| \le \alpha_2$.

**Assumption 1.** $F(.)$ *is continuous and* $\frac{\partial F(t, x, u_{fs})}{\partial u_{fs}}$ *is bounded.*

To facilitate the design of the controller, a Taylor series expansion is applied to convert the control input in the non-affine nonlinear system of Equation (4) to an explicit expression, resulting in

$$\begin{cases} \dot{x}_1 = x_2 \\ \dot{x}_2 = f(t, x) + g(t, x)\rho\, u_s + d \end{cases} \tag{5}$$

where $f(t, x) = F(t, x, u_{fs_\xi}) - \frac{\partial F(t, x, u_{fs})}{\partial u_{fs}}|_{u_{fs\xi}} u_{fs\xi}$, $g = \frac{\partial F(t, x, u_{fs})}{\partial u_{fs}}|_{u_{fs\xi}}$, $d = D + \Delta(.)$ is the compound disturbance, $\Delta(.)$ is a higher-order term, and $u_{fs\xi}$ denotes the filtered value of $u_{fs}$ which is acquired by a robust sliding mode differentiator.

$$\dot{u}_{fs\xi} = -\frac{u_{fs\xi} - u_{fs}}{\tau} - \frac{\zeta_1(u_{fs\xi} - u_{fs})}{\left\| u_{fs\xi} - u_{fs} \right\| + \zeta_2} \tag{6}$$

where $\tau$ is the filter time constant and the two positive parameters $\zeta_1$ and $\zeta_2$ denote the switching gain and the switching rate to regulate the sliding mode, respectively.

**Remark 1.** *From Equation (6), $u_{fs\xi}$ can approximate $u_{fs}$ with any small deviation by choosing an appropriate the filter time parameter $\tau$. If present, the higher-order term $\Delta(.)$ can tend to zero. Supposing that the $\Delta(.)$ is small enough, its effect can be compensated in d.*

To proceed to the following work, the definitions and lemmas are given.

**Definition 1.** *[31] If on an open region $\mathcal{D}$ containing the origin defined about the system $\dot{x} = f(x)$, a scalar function $V(x)$ that is continuously differentiable and positive definite has the property $V(x) \to \infty$ as x approaches the boundary of $\mathcal{D}$ and has positive constant boundedness along with the solution of the system $\dot{x} = f(x)$ with $x(0) \in \mathcal{D}$, then $V(x)$ is known as a barrier Lyapunov function.*

In brief, the symmetric barrier Lyapunov function is defined as

$$V_i = \frac{1}{2}\ln\left(\frac{k_{ai}^2}{k_{ai}^2 - z_i^2}\right) \tag{7}$$

If $z_i$ stays within the boundary of $k_{ai}$, especially since the barrier Lyapunov function tends toward infinity at $|z_i| = k_{ai}$, there can exist an available Lyapunov function in the interval $|z_i| < k_{ai}$.

**Lemma 1.** [31] *For any positive constant $k_{ai} \in R$, the following equality satisfies $z_i \in R$ in the set $|z_i| < k_{ai}$:*

$$\ln\frac{k_{ai}^2}{k_{ai}^2 - z_i^2} \leq \frac{z_i^2}{k_{ai}^2 - z_i^2} \tag{8}$$

**Lemma 2.** *(Young's inequality) For $\forall\ x, y \geq 0$, the following inequality holds:*

$$xy \leq \frac{\epsilon^p}{p}x^p + \frac{1}{q\epsilon^q}y^q \tag{9}$$

*with $\epsilon > 0, p > 1, q > 1, \frac{1}{p} + \frac{1}{q} = 1$. Only if $x^p = y^q$, the equals sign holds in Equation (9).*

**Lemma 3.** *For any positive constant $\varepsilon > 0$ and $x \in R$, the following is satisfied:*

$$0 \leq |x| - x\tanh(\varepsilon x) \leq \iota/\varepsilon \tag{10}$$

*where the normal number $\iota = e^{-(\iota+1)}$, namely $\iota \approx 0.2785$.*

**Lemma 4.** [40] *The first order sliding mode differentiator is expressed as*

$$\begin{cases} \dot{\omega}_1 = \gamma_0 = -\epsilon_1|\omega_1 - f(t)|^{\frac{1}{2}}sign(\omega_1 - f(t)) + \omega_2 \\ \dot{\omega}_2 = \epsilon_2 sign(\omega_2 - \gamma_0) \end{cases} \tag{11}$$

*where $\omega_1, \gamma_0$, and $\omega_2$ denote states of the first order sliding mode differentiator, $\epsilon_1$ and $\epsilon_2$ are the designed parameters, and $f(t)$ is an unknown function. Consequently, $\gamma_0$ can approximate the differential term $\dot{f}(t)$ with any arbitrary precision if the initial errors $\omega_1(t_0) - f(t_0), \omega_2(t_0) - \dot{f}(t_0)$ are bounded.*

**Lemma 5.** [41] *If there exists a continuously positive function $V(x, t) : R^n \times R^+ \to R^+$, with two scalars $c_1 > 0$ and $c_2 \geq 0$ and where $\mu_1$ and $\mu_2$ are class $K_\infty$-functions, then $V(x, t)$ satisfies Equations (12) and (13)*

$$\mu_1(|x|) \leq V(x, t) \leq \mu_2(|x|) \tag{12}$$

$$\dot{V} \leq -c_1 V + c_2 \tag{13}$$

*with regard to $x \in R^n$ and $t > 0$. Then, there is the following solution for arbitrarily initial value $x(0) \in R^n$ and satisfies*

$$V(x, t) \leq V(0)e^{-c_1 t} + \frac{c_2}{c_1}, \quad \forall t > 0 \tag{14}$$

**Lemma 6.** [42] *Let the unknown function $f(Z)$ be defined over a compact set $\Omega_z$; then for any approximation accuracy $l^* > 0$, there exists an RBFNN such that*

$$f(Z) = W^{*T}\Psi(Z) + l(Z) \tag{15}$$

*where $Z \in \Omega_z \subset R^n$ is the input vector of the neural networks with n being the input dimension. $W = [w_1, w_2, \cdots, w_m]^T \in R^m$ is the weight vector with the neural network node number m and $W^*$ is the ideal*

constant weight vector. $l(Z)$ denotes the approximation error with $\|l(Z)\| \leq l^*$. $\Psi(Z)$ is the smooth basis vector $\Psi(Z) = [\Psi_1(Z), \Psi_2(Z), \cdots, \Psi_m(Z)]^T \in \mathbf{R}^m$ with $\Psi_i(Z)$ selected from the commonly used Gaussian functions

$$\Psi_i(Z) = exp\left[\frac{-(Z - c_i)^T(Z - c_i)}{b_i^2}\right], \quad i = 1, 2, \cdots, m \tag{16}$$

where $c_i = [c_{i1}, c_{i2}, \cdots, c_{in}]^T$ and $b_i$ are the center and width of the Gaussian functions, respectively. The ideal constant weight vector is defined such as

$$W^* := arg \min_{\hat{W} \in \mathbf{R}^m} \left\{ \sup_{Z \in \Omega_z} |f(Z) - \hat{W}^T\Psi(Z)|) \right\} \tag{17}$$

It's noteworthy that the ideal weight vector $W^*$ is unknown and its elements need to be estimated by designed adaptive law. Nevertheless, in this paper, minimum parameter learning theory is used instead of a direct estimate with the variable $\theta = \bar{G}^{-1}\|W^*\|^2$.

**Remark 2.** *The ideal weight $W^* \in \mathbf{R}^m$ realizes $m$ unknown elements to be estimated. More estimators will result in a good many of adjustment parameters, thus directly increasing the computational burden. By updating the estimation values of the norm for the unknown neural network weight vectors but not their weights only the unknown parameter $\theta$ needs to be estimated in this paper, hence the computational complexity can be reduced substantially.*

For the system of Equation (4), it is necessary for the system transformation and the reference trajectory to satisfy the following assumptions:

**Assumption 2.** *There exist two positive constants $\beta_1$ and $\beta_2$ such that the desired reference trajectory signal $y_d$ and its time derivative $\dot{y}_d$ have corresponding boundedness, i.e., $|y_d| \leq \beta_1$, $|\dot{y}_d| < \beta_2$.*

**Assumption 3.** *For a time-varying unknown compound disturbance $d$, there exists an unknown positive constant $d_m$, i.e., $|d| < d_m$.*

**Assumption 4.** *Without loss of generality, there exist two negative constants $\underline{G} < \bar{G} < 0$ such that $\underline{G} < g\rho < \bar{G} < 0$.*

**Remark 3.** *This assumption is widely used as a necessary controllable condition in [43] which implies that $g$ is strictly negative and that the affine nonlinear system is nonsingular. Since the bounds of $\underline{G}$ and $\bar{G}$ need not be known, the assumption has a broad scope of application.*

In this paper, an adaptive neural network fault-tolerant control scheme is investigated for a SISO non-affine nonlinear yaw control system with provisions for the composite factor of actuator faults, input saturation, full-state constraints, and external disturbances. First of all, the non-affine nonlinear system is converted into an affine nonlinear expression via an integrating Taylor series expansion with a robust sliding mode differentiator. Next, an RBFNN is applied to approximate the unknown function, and an adaptive control design is employed to deal with the compound bounded disturbance and the approximation error of the RBFNN. Then, an anti-saturation compensator is used to avoid the influence of input saturation on the performance of the closed-loop system. Lastly, a controller based on the barrier Lyapunov function is designed to guarantee good tracking control performance. The control structure is shown in Figure 1.

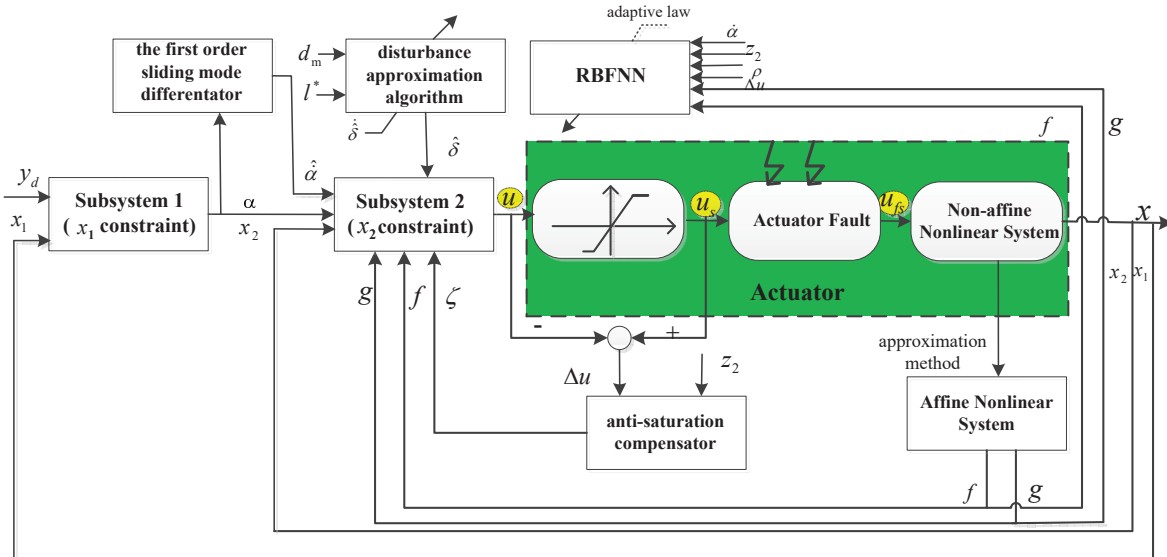

**Figure 1.** The adaptive neural fault-tolerant control structure for UAV helicopter yaw control system with input saturation and full-state constraints.

### 2.3. Controller Design and Stability Analysis

In this section, we present the adaptive neural network fault-tolerant control using the backstepping technique for helicopter yaw control system. The recursive design process contains two steps in the system of Equation (5).The design procedure is as follows:

Step 1: The tracking error is defined as $z_1 = x_1 - y_d$ and its time derivative is

$$\dot{z}_1 = \dot{x}_1 - \dot{y}_d = x_2 - \dot{y}_d \tag{18}$$

Choose the barrier Lyapunov function candidate as

$$V_1 = \frac{1}{2} \ln \frac{k_{a1}^2}{k_{a1}^2 - z_1^2} \tag{19}$$

where $k_{a1}$ is a positive scalar. By introducing $z_2 = x_2 - \alpha$, where $\alpha$ is the designed virtual control signal to be defined later, the derivative of $V_1$ is given by

$$\dot{V}_1 = \frac{z_1 \dot{z}_1}{k_{a1}^2 - z_1^2} = \frac{z_1 (z_2 + \alpha - \dot{y}_d)}{k_{a1}^2 - z_1^2} \tag{20}$$

Design the virtual control $\alpha$ as

$$\alpha = -k_1 z_1 + \dot{y}_d - \frac{z_1}{2(k_{a1}^2 - z_1^2)} \tag{21}$$

with $k_1 > 0$. Based on Equations (20) and (21), then, one has

$$\dot{V}_1 = \frac{-k_1 z_1^2}{k_{a1}^2 - z_1^2} - \frac{z_1^2}{2(k_{a1}^2 - z_1^2)^2} + \frac{z_1 z_2}{k_{a1}^2 - z_1^2} \tag{22}$$

Then from Lemma 2, we obtain

$$\frac{z_1 z_2}{k_{a1}^2 - z_1^2} \le \frac{z_1^2}{2(k_{a1}^2 - z_1^2)^2} + \frac{1}{2} z_2^2 \tag{23}$$

By substituting (23) into (22), we have

$$\dot{V}_1 \le \frac{-k_1 z_1^2}{k_{a1}^2 - z_1^2} + \frac{1}{2} z_2^2 \tag{24}$$

Step 2: In this step, the actual control input will be derived. Selecting the barrier Lyapunov function as

$$V_2 = V_1 + \frac{1}{2} \ln \frac{k_{a2}^2}{k_{a2}^2 - z_2^2} \tag{25}$$

where $k_{a2}$ is a positive number, we define the variable $\Delta u = u_s - u$. The time derivative of the error variable $z_2$ is

$$\begin{aligned}
\dot{z}_2 &= \dot{x}_2 - \dot{\alpha} \\
&= f + g\rho u_s + d - \dot{\alpha} \\
&= f + g\rho \Delta u + g\rho u - \dot{\alpha} + d
\end{aligned} \tag{26}$$

Invoking (24), (25), and (26),

$$\begin{aligned}
\dot{V}_2 &= \dot{V}_1 + \frac{z_2 \dot{z}_2}{k_{a2}^2 - z_2^2} \\
&= \dot{V}_1 + \frac{z_2}{k_{a2}^2 - z_2^2} (f + g\rho \Delta u + g\rho u - \dot{\alpha} + d) \\
&\le \frac{-k_1 z_1^2}{k_{a1}^2 - z_1^2} + \frac{z_2}{k_{a2}^2 - z_2^2} \left[ f + g\rho \Delta u + g\rho u - \dot{\alpha} + d + \frac{z_2}{2}(k_{a2}^2 - z_2^2) \right]
\end{aligned} \tag{27}$$

where $\dot{\alpha}$ can be estimated by Lemma 4.

The unknown nonlinear function is defined as $F(Z) = f + g\rho \Delta u - \dot{\alpha} + \frac{z_2}{2}(k_{a2}^2 - z_2^2)$. The RBFNN is used to approximate the unknown function $F(Z)$ as

$$F(Z) = W^{*T} \Psi(Z) + l(Z), \quad |l(Z)| \le l^* \tag{28}$$

where $Z = [x_1, x_2, y_d, \dot{y}_d]^T \in \Omega_Z$, $l(Z)$ is the approximation error. To reduce the computational burden, the variable $\theta = \bar{G}^{-1} \|W^*\|^2$ is applied.

Substituting (28) into (27), we have

$$\dot{V}_2 \le \frac{-k_1 z_1^2}{k_{a1}^2 - z_1^2} + \frac{z_2}{k_{a2}^2 - z_2^2} W^{*T} \Psi + \frac{z_2}{k_{a2}^2 - z_2^2}(d + l) + \frac{z_2}{k_{a2}^2 - z_2^2} g\rho u \tag{29}$$

By using Young's inequality, the second term of Equation (29) can be rewritten as

$$\frac{z_2}{k_{a2}^2 - z_2^2} W^{*T} \Psi \le \frac{z_2^2}{2a^2(k_{a2}^2 - z_2^2)^2} \|W^*\|^2 \Psi^T \Psi + \frac{a^2}{2} \tag{30}$$

where $a$ is a positive design parameter.

A new variable is defined as $\delta = \bar{G}^{-1}(d_m + l^*)$ and based on Assumption 3, we introduce $\delta$, $\theta$ and (30) into (29) and obtain

$$\begin{aligned}
\dot{V}_2 &\le \frac{-k_1 z_1^2}{k_{a1}^2 - z_1^2} + \frac{z_2}{k_{a2}^2 - z_2^2} g\rho u + \frac{z_2^2}{2a^2(k_{a2}^2 - z_2^2)^2} \|W^*\|^2 \Psi^T \Psi + \frac{|z_2|}{k_{a2}^2 - z_2^2} \bar{G}\delta + \frac{a^2}{2} \\
&= \frac{-k_1 z_1^2}{k_{a1}^2 - z_1^2} + \frac{z_2}{k_{a2}^2 - z_2^2} g\rho u + \frac{z_2^2}{2a^2(k_{a2}^2 - z_2^2)^2} \bar{G}\theta \Psi^T \Psi + \frac{|z_2|}{k_{a2}^2 - z_2^2} \bar{G}\delta + \frac{a^2}{2}
\end{aligned} \tag{31}$$

The adaptive law for the unknown parameters $\theta$ and $\delta$ is given by

$$
\begin{cases}
\dot{\hat{\theta}} = \gamma \left[ \sigma_1 \hat{\theta} - \dfrac{z_2{}^2}{2a^2(k_{a2}^2 - z_2^2)^2} \Psi^T \Psi \right] \\[4mm]
\dot{\hat{\delta}} = \beta \left[ \sigma_2 \hat{\delta} - \dfrac{z_2}{k_{a2}^2 - z_2^2} tanh(\dfrac{z_2}{\lambda(k_{a2}^2 - z_2^2)}) \right]
\end{cases}
\tag{32}
$$

where $\hat{\theta}$ and $\hat{\delta}$ are the estimated values of the unknown parameters $\theta$ and $\delta$, respectively. $\gamma > 0, \beta > 0$, $\lambda > 0$, $\sigma_1 < 0$ and $\sigma_2 < 0$ are parameters to be designed. The estimated errors can be expressed as $\tilde{\theta} = \theta - \hat{\theta}$ and $\tilde{\delta} = \delta - \hat{\delta}$. The initial values of $\hat{\theta}$ and $\hat{\delta}$ are assumed to be satisfied: $\hat{\theta}(0) \leq 0$ and $\hat{\delta}(0) \leq 0$ according to [44], which will realize these conditions for $t > 0$.

To reduce the problem of control input saturation, the anti-saturation compensator is designed as follows:

$$
\dot{\zeta} = -(k_3 + \frac{k_4}{k_{a2}^2 - z_2^2})\zeta + \Delta u
\tag{33}
$$

where $\zeta$ denotes the state of the compensator with $k_4 > k_3 > 0$.

Step 3: The barrier Lyapunov function candidate $V$ is chosen as

$$
V = V_2 - \frac{\bar{G}\tilde{\delta}^2}{2\beta} - \frac{\bar{G}\tilde{\theta}^2}{2\gamma} + \frac{1}{2}\zeta^2
\tag{34}
$$

The time derivative of $V$ is

$$
\begin{aligned}
\dot{V} &= \dot{V}_2 + \frac{\bar{G}}{\beta}\tilde{\delta}\dot{\hat{\delta}} + \frac{\bar{G}}{\gamma}\tilde{\theta}\dot{\hat{\theta}} + \zeta\dot{\zeta} \\[2mm]
&\leq \frac{-k_1 z_1^2}{k_{a1}^2 - z_1^2} + \frac{z_2}{k_{a2}^2 - z_2^2}g\rho u + \frac{z_2{}^2}{2a^2(k_{a2}^2 - z_2^2)^2}\bar{G}\theta\Psi^T\Psi + \frac{|z_2|}{k_{a2}^2 - z_2^2}\bar{G}\delta + \frac{a^2}{2} \\[2mm]
&\quad + \frac{\bar{G}}{\beta}\tilde{\delta}\dot{\hat{\delta}} + \frac{\bar{G}}{\gamma}\tilde{\theta}\dot{\hat{\theta}} + \zeta\dot{\zeta} \\[2mm]
&= \frac{-k_1 z_1^2}{k_{a1}^2 - z_1^2} + \frac{z_2}{k_{a2}^2 - z_2^2}g\rho u + \frac{z_2{}^2}{2a^2(k_{a2}^2 - z_2^2)^2}\bar{G}\theta\Psi^T\Psi + \frac{|z_2|}{k_{a2}^2 - z_2^2}\bar{G}\delta + \frac{a^2}{2} \\[2mm]
&\quad + \bar{G}\tilde{\delta}\sigma_2\hat{\delta} - \bar{G}\tilde{\delta}\frac{z_2}{k_{a2}^2 - z_2^2}tanh(\frac{z_2}{\lambda(k_{a2}^2 - z_2^2)}) + \bar{G}\tilde{\theta}\sigma_1\hat{\theta} - \bar{G}\tilde{\theta}\frac{z_2{}^2}{2a^2(k_{a2}^2 - z_2^2)^2}\Psi^T\Psi \\[2mm]
&\quad - k_3\zeta^2 - \frac{k_4}{k_{a2}^2 - z_2^2}\zeta^2 + \Delta u\zeta
\end{aligned}
\tag{35}
$$

Now, the actual control input $u$ is designed as

$$
u = k_2 z_2 - k_2 \zeta - \frac{\hat{\theta}z_2}{2a^2(k_{a2}^2 - z_2^2)}\Psi^T\Psi - \hat{\delta}tanh(\frac{z_2}{\lambda(k_{a2}^2 - z_2^2)})
\tag{36}
$$

where $k_2$ is a positive design parameter. Considering $u$ into the second term of Equation (35), we get

$$
\begin{aligned}
\frac{z_2}{k_{a2}^2 - z_2^2}g\rho u &= \frac{z_2}{k_{a2}^2 - z_2^2}g\rho \left[ k_2 z_2 - k_2 \zeta - \frac{\hat{\theta}z_2}{2a^2(k_{a2}^2 - z_2^2)}\Psi^T\Psi - \hat{\delta}tanh(\frac{z_2}{\lambda(k_{a2}^2 - z_2^2)}) \right] \\[2mm]
&\leq \frac{k_2 z_2{}^2}{k_{a2}^2 - z_2^2}\bar{G} - \frac{k_2 z_2 \zeta}{k_{a2}^2 - z_2^2}g\rho - \frac{\hat{\theta}z_2{}^2}{2a^2(k_{a2}^2 - z_2^2)^2}\Psi^T\Psi\bar{G} - \frac{z_2}{k_{a2}^2 - z_2^2}\hat{\delta}tanh(\frac{z_2}{\lambda(k_{a2}^2 - z_2^2)})\bar{G}
\end{aligned}
\tag{37}
$$

By applying Young's inequality, the following inequalities hold:

$$-\frac{k_2 z_2 \zeta}{k_{a2}^2 - z_2^2} g\rho \leq \frac{k_2}{k_{a2}^2 - z_2^2}|z_2||\zeta||g\rho|$$

$$\leq -\frac{k_2 z_2^2 \bar{G}}{2(k_{a2}^2 - z_2^2)} - \frac{k_2 \zeta^2 \underline{G}^2}{2(k_{a2}^2 - z_2^2)\bar{G}}$$

$$\Delta u \zeta \leq \frac{1}{2k_3}\Delta u^2 + \frac{k_3}{2}\zeta^2$$

(38)

From (37) and (38), the time derivative of $V$ is expressed as

$$
\begin{aligned}
\dot{V} \leq & -\frac{k_1 z_1^2}{k_{a1}^2 - z_1^2} + \frac{k_2 z_2^2}{k_{a2}^2 - z_2^2}\bar{G} - \frac{k_2 z_2^2}{2(k_{a2}^2 - z_2^2)}\bar{G} - \frac{k_2 \zeta^2 \underline{G}^2}{2(k_{a2}^2 - z_2^2)\bar{G}} - \frac{\hat{\theta} z_2^2}{2a^2(k_{a2}^2 - z_2^2)^2}\Psi^T \Psi \bar{G} \\
& - \frac{z_2}{k_{a2}^2 - z_2^2}\hat{\delta}tanh(\frac{z_2}{\lambda(k_{a2}^2 - z_2^2)})\bar{G} + \frac{z_2^2}{2a^2(k_{a2}^2 - z_2^2)^2}\bar{G}\theta\Psi^T\Psi + \frac{|z_2|}{k_{a2}^2 - z_2^2}\bar{G}\delta + \frac{a^2}{2} \\
& + \bar{G}\tilde{\delta}\sigma_2\hat{\delta} - \bar{G}\tilde{\delta}\frac{z_2}{k_{a2}^2 - z_2^2}tanh(\frac{z_2}{\lambda(k_{a2}^2 - z_2^2)}) + \bar{G}\tilde{\theta}\sigma_1\hat{\theta} - \bar{G}\tilde{\theta}\frac{z_2^2}{2a^2(k_{a2}^2 - z_2^2)^2}\Psi^T\Psi \\
& - k_3\zeta^2 - \frac{k_4}{k_{a2}^2 - z_2^2}\zeta^2 + \frac{1}{2k_3}\Delta u^2 + \frac{k_3}{2}\zeta^2 \\
= & -\frac{k_1 z_1^2}{k_{a1}^2 - z_1^2} + \frac{k_2 z_2^2}{2(k_{a2}^2 - z_2^2)}\bar{G} - \frac{\zeta^2}{k_{a2}^2 - z_2^2}(k_4 + \frac{k_2\underline{G}^2}{2\bar{G}}) + \bar{G}\tilde{\delta}\sigma_2\hat{\delta} + \bar{G}\tilde{\theta}\sigma_1\hat{\theta} + \frac{a^2}{2} \\
& - \frac{k_3}{2}\zeta^2 + \frac{1}{2k_3}\Delta u^2 + \bar{G}\delta\left[\frac{|z_2|}{k_{a2}^2 - z_2^2} - \frac{z_2}{k_{a2}^2 - z_2^2}tanh(\frac{z_2}{\lambda(k_{a2}^2 - z_2^2)})\right] \\
\leq & -\frac{k_1 z_1^2}{k_{a1}^2 - z_1^2} - \frac{k_2 z_2^2}{2(k_{a2}^2 - z_2^2)}(-\bar{G}) - \frac{k_3}{2}\zeta^2 + \bar{G}\tilde{\delta}\sigma_2\hat{\delta} + \bar{G}\tilde{\theta}\sigma_1\hat{\theta} + \frac{a^2}{2} + \frac{\Delta u^2}{2k_3} + 0.2785\lambda\bar{G}\delta
\end{aligned}
$$

(39)

Since $\tilde{\delta} = \delta - \hat{\delta}$ and $\tilde{\theta} = \theta - \hat{\theta}$, the following inequalities hold:

$$\tilde{\delta}\hat{\delta} = \tilde{\delta}(\delta - \tilde{\delta}) \leq -\frac{1}{2}\tilde{\delta}^2 + \frac{1}{2}\delta^2$$

$$\tilde{\theta}\hat{\theta} = \tilde{\theta}(\theta - \tilde{\theta}) \leq -\frac{1}{2}\tilde{\theta}^2 + \frac{1}{2}\theta^2$$

(40)

Considering (40) and Lemma 1, we have

$$
\begin{aligned}
\dot{V} \leq & -\frac{k_1 z_1^2}{k_{a1}^2 - z_1^2} - \frac{k_2 z_2^2}{2(k_{a2}^2 - z_2^2)}(-\bar{G}) - \frac{\bar{G}\sigma_2 \tilde{\delta}^2}{2} - \frac{\bar{G}\sigma_1 \tilde{\theta}^2}{2} - \frac{k_3}{2}\zeta^2 \\
& + \frac{a^2}{2} + \frac{\Delta u^2}{2k_3} + 0.2785\lambda\bar{G}\delta + \frac{\bar{G}\sigma_2}{2}\delta^2 + \frac{\bar{G}\sigma_1}{2}\theta^2 \\
\leq & -k_1 \ln\frac{k_{a1}^2}{k_{a1}^2 - z_1^2} - \frac{-k_2\bar{G}}{2}\frac{k_{a2}^2}{k_{a2}^2 - z_2^2} - \frac{\bar{G}\sigma_2 \tilde{\delta}^2}{2} - \frac{\bar{G}\sigma_1 \tilde{\theta}^2}{2} - \frac{k_3}{2}\zeta^2 \\
& + \frac{a^2}{2} + \frac{\Delta u^2}{2k_3} + 0.2785\lambda\bar{G}\delta + \frac{\bar{G}\sigma_2}{2}\delta^2 + \frac{\bar{G}\sigma_1}{2}\theta^2
\end{aligned}
$$

(41)

It should be noted that the state $\zeta$ of the anti-saturation compensator is affected by the value of $|\Delta u|$. If $|\Delta u|$ tends to infinity, the control input will be infinite when the system (5) tracks the reference trajectory which will cause the saturation compensator to cease to be effective; the system will not track the desired value. Based on this analysis, $|\Delta u|$ should be bounded.

**Remark 4.** *In this paper, an anti-saturation compensator (33) is used to avoid the problem of input saturation. When the control input required by the system is greater than the actuator performance, $\Delta u \neq 0$. At this time, the state $\zeta$ is generated by the anti-saturation compensator, which compensates for the deviation caused by input saturation until $\Delta u = 0$. Furthermore, the larger the values of parameters $k_3$ and $k_4$ are, the faster the auxiliary system's compensation rate is. Therefore, with the anti-saturation auxiliary system, this paper's design effectively deals with the adverse effects created by input saturation and guarantees helicopter's yaw angle true tracking to the desired trajectory under the input constraints.*

**Theorem 1.** *Consider the non-affine nonlinear system (4), affine-form system (5), and Assumptions 1–4. By establishing an adaptive saturation controller (36), a virtual control signal (21), and by designing the adaptation law (32), with the initial conditions bounded and all signals of the closed-loop system uniformly bounded, the proposed scheme can guarantee that the helicopter's yaw channel system tracking errors $z_1$ and $z_2$ converge to small compact sets around the origin.*

**Proof.** We can get the following inequality from (41):

$$\dot{V} \leq -kV + C \tag{42}$$

where $k = min\{2k_1, -k_2\bar{G}, -\beta\sigma_2, -\gamma\sigma_1, k_3\}$ and $C = \frac{a^2}{2} + \frac{\Delta u^2}{2k_3} + 0.2785\lambda\bar{G}\delta + \frac{\bar{G}\sigma_2}{2}\delta^2 + \frac{\bar{G}\sigma_1}{2}\theta^2$.

At the same time, integrating both sides of (42) over $[0, t]$, we get

$$0 \leq V(t) \leq \left[V(0) - \frac{C}{k}\right]e^{-kt} + \frac{C}{k} \tag{43}$$

where $V(0)$ is the initial value of $V$. Equation (43) shows that $z_1, z_2, \tilde{\delta}, \tilde{\theta}$, and $\zeta$ are bounded.

Then, because $x_1 = z_1 + y_d$ and $|y_d| \leq \beta_1$, we have $|x_1| \leq |z_1| + |y_d| < k_{a1} + \beta_1 \leq \alpha_1$. From the design of $\alpha$ in (21), we know the function $\alpha$ contains $x_1, y_d$ and $\dot{y}_d$, and that the maximum value of $\alpha$ exists, namely $\alpha \leq \alpha_m$. Considering that $x_2 = z_2 + \alpha$, we get $|x_2| \leq |z_2| + |\alpha| < k_{a2} + \alpha_m \leq \alpha_2$. Since $\delta, \tilde{\delta}, \theta$, and $\tilde{\theta}$ are bounded, $\hat{\delta} = \delta - \tilde{\delta}$ and $\hat{\theta} = \theta - \tilde{\theta}$ are also bounded. The designed actual control input $u$ from (36) verifies that $u$ is bounded. Therefore, all the signals in the closed-loop system remain uniformly bounded.

Further analysis shows that

$$\ln \frac{k_{a1}^2}{k_{a1}^2 - z_1^2} \leq 2[V(0) - \frac{C}{k}]e^{-kt} + \frac{2C}{k}$$
$$\ln \frac{k_{a2}^2}{k_{a2}^2 - z_2^2} \leq 2[V(0) - \frac{C}{k}]e^{-kt} + \frac{2C}{k} \tag{44}$$

Taking exponentials on both sides of (44) leads to

$$\frac{k_{a1}^2}{k_{a1}^2 - z_1^2} \leq e^{2[V(0) - \frac{C}{k}]e^{-kt} + \frac{2C}{k}}$$
$$\frac{k_{a2}^2}{k_{a2}^2 - z_2^2} \leq e^{2[V(0) - \frac{C}{k}]e^{-kt} + \frac{2C}{k}} \tag{45}$$

Since $k_{a1}^2 - z_1^2 > 0$ and $k_{a2}^2 - z_2^2 > 0$, multiplying by $k_{a1}^2 - z_1^2$ and $k_{a2}^2 - z_2^2$ in (45) results in

$$|z_1| \leq k_{a1}\sqrt{1 - e^{-2[V(0) - \frac{C}{k}]e^{-kt} - \frac{2C}{k}}}$$
$$|z_2| \leq k_{a2}\sqrt{1 - e^{-2[V(0) - \frac{C}{k}]e^{-kt} - \frac{2C}{k}}} \tag{46}$$

It can be seen that given any $\Delta_1 > k_{a1}\sqrt{1 - e^{-2\frac{C}{k}}}$ and $\Delta_2 > k_{a2}\sqrt{1 - e^{-2\frac{C}{k}}}$, there is a $T$ such that for any $t > T$, $|z_1| \le \Delta_1$ and $|z_2| \le \Delta_2$. $|z_1| \le k_{a1}\sqrt{1 - e^{-2\frac{C}{k}}}$ and $|z_2| \le k_{a2}\sqrt{1 - e^{-2\frac{C}{k}}}$ along with $t \to \infty$. It can be concluded that $|z_1| \le k_{a1}\sqrt{1 - e^{-2\frac{C}{k}}}$ and $|z_2| \le k_{a2}\sqrt{1 - e^{-2\frac{C}{k}}}$ as $t \to \infty$. According to the definitions of $C$ and $k$ in (42), we conclude that $z_1$ and $z_2$ eventually converge to arbitrarily small compact sets by choosing appropriate design parameters. The proof is completed. □

**Remark 5.** *In this paper, the barrier Lyapunov theory integrated with the design procedure of backstepping is applied to deal with state constraints. The change range of backstepping error variables $z_1$ and $z_2$ is restricted by the design parameters $k_{a1}$ and $k_{a2}$. If $|z_1| \to k_{a1}$ and $|z_2| \to k_{a2}$, the barrier Lyapunov function will approximate infinity. Hence, the variation of $z_1$ and $z_2$ is always limited to $|z_1| < k_{a1}$ and $|z_2| < k_{a2}$. This ensures that the bounds of the system state constraints are not violated in the process of helicopter's yaw channel tracking, satisfying the limits of helicopter yaw motion space and improving its operational security.*

**Remark 6.** *The investigated control strategy is an adaptive neural fault-tolerant control, and it is employed in a UAV helicopter SISO non-affine nonlinear yaw control system to realize tracking errors in arbitrarily small compact sets; As far as we know, there are few conclusions about adaptive neural network fault-tolerant control for non-affine nonlinear yaw control system faced with the uncertainties in system conversion, unknown disturbances, actuator faults, input saturations, and full-state constraints. The MIMO systems can also make use of this control scheme, but the universality of the proposed controller needs further study.*

*2.4. Simulation Results*

A UAV yaw-channel model acquired from the helicopter-on-arm platform [45,46] was used to confirm feasibility of the proposed control scheme.Taking input saturation and actuator faults into consideration for the UAV yaw-channel model, the non-affine nonlinear yaw dynamic model was formulated as follows:

$$\begin{cases} \dot{\varphi} = r \\ \dot{r} = l_1\varphi + l_2 r + l_3\theta_{trfs} + l_4\theta_{trfs}^2 + l_5\Omega\theta_{trfs} + D \end{cases} \tag{47}$$

with $l_1 = -3.33$, $l_2 = -1.38$, $l_3 = 63.09$, $l_4 = 11.65$, $l_5 = -0.14$, and $\Omega = 1200$. $\theta_{trfs}$ denotes the pitch angle with saturation and actuator faults. $D = \sin t + \cos 2t + 2$ is an external disturbance.

Let $[\varphi, r]^T = [x_1, x_2]^T$, $\theta_{trfs} = u_{fs}$. The nonlinear controller was designed by applying the Taylor series expansion technique to Equation (47).

$$\begin{cases} \dot{x}_1 = x_2 \\ \dot{x}_2 = f + g\rho u_s + d \end{cases} \tag{48}$$

where $f = l_1 x_1 + l_2 x_2 + l_3 u_{fs\zeta} + l_4 u_{fs\zeta}^2 + l_5\Omega u_{fs\zeta} - (l_3 + 2l_4 u_{fs\zeta} + l_5\Omega)u_{fs\zeta}$, $g = l_3 + 2l_4 u_{fs\zeta} + l_5\Omega$.

The initial conditions of the UAV helicopter are $x_1(0) = 0.16$ rad and $x_2(0) = -0.2$ rad/s. The desired tracking command was $y_d = 0.52\sin(1.2t) + 0.05\cos(t) + 0.05$. With $k_{a1} = k_{a2} = 0.2$, the state constraints of the system were $|x_1| \le 0.82$ rad, $|x_2| \le 0.874$ rad/s. The robust sliding mode filter parameters in Equation (6) were chosen as $\tau = 0.005$, $\zeta_1 = 20$, and $\zeta_2 = 0.01$. The saturation parameter $u_{sm} = 0.055$ rad, the anti-saturation compensator parameters were designed as $k_3 = 0.5$, $k_4 = 1$, and the initial value of $\zeta(0) = 0$.

The RBFNN was exerted to approximate the unknown nonlinear function. The neural network $W^{*T}\Psi(Z)$ contained six units evenly distributed in the interval $[-25, 25]$ with the width of each unit equal to 9. The designed adaptive parameters were taken as $\lambda = 0.2$, $\gamma = \beta = 1$, $\sigma_1 = -0.3$, $\sigma_2 = -0.1$, and $a = 0.1$. The initial estimated values of the adaptive parameters were $\hat{\theta}(0) = 0$ and $\hat{\delta}(0) = -0.1$. The input controller parameters were chosen as $k_1 = 0.5$ and $k_2 = 1$. In the simulation scenario, the

UAV helicopter suffers from actuator fault at $t \geq 15s$, and $\rho = \frac{1}{0.75+e^{-cosx_1x_2}}$. To verify the feasibility of the proposed adaptive neural fault-tolerant control scheme, the simulation results are shown in Figures 2–9. Figure 2 shows the approximation curves of the states which illustrate the effectiveness of the approximation method (5). Figure 3 shows the good approximation performance of the robust sliding mode filter (6); the filtered value $u_{fs\zeta}$ can approximate $u_{fs}$ with good accuracy. Figures 4 and 5 not only demonstrate the good tracking performance of system states with input saturation and an actuator fault but also meet state constraints. Figures 6 and 7 give the time trajectories of the adaptive parameters $\hat{\theta}$ and $\hat{\delta}$. Figure 8 shows the state $\zeta$ change of the anti-saturation compensator. Figure 9 shows the changes of helicopter input pitch angle $u$ and input with saturation and actuator fault $u_{fs}$. According to the simulation, the anti-saturation compensator can accommodate input constraints and demonstrates a superior compensation characteristic. The adaptive saturation controller of Equation (36) proposed in this paper can effectively address the state constraints and input saturation of the system. We can see that when an actuator fault occurs at $t \geq 15$ s, the system can still track the desired trajectory.

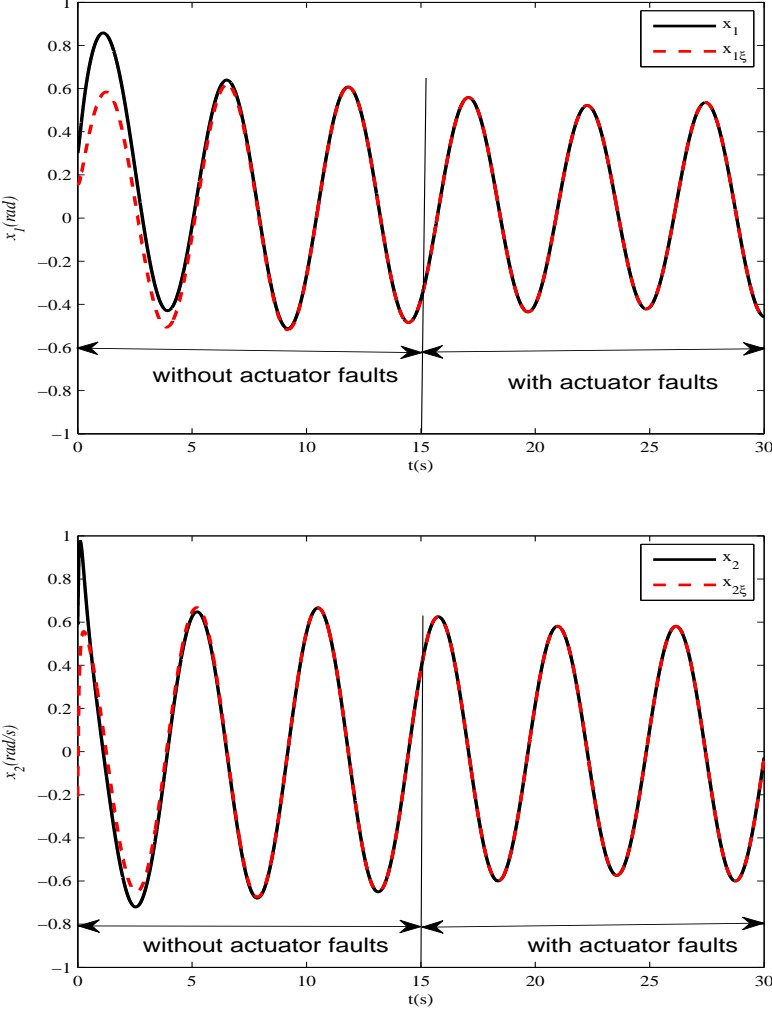

**Figure 2.** The response curves of $x_1$ and $x_2$ in system transformation.

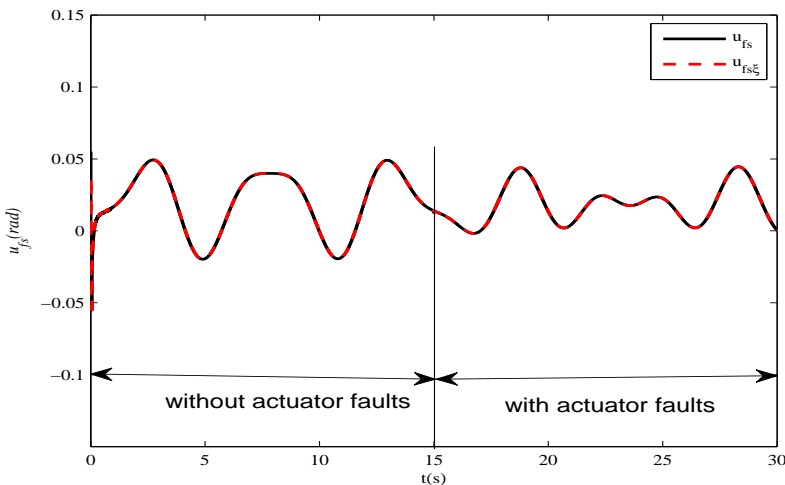

**Figure 3.** The approximation performance of $u_{f_s}$ in system transformation.

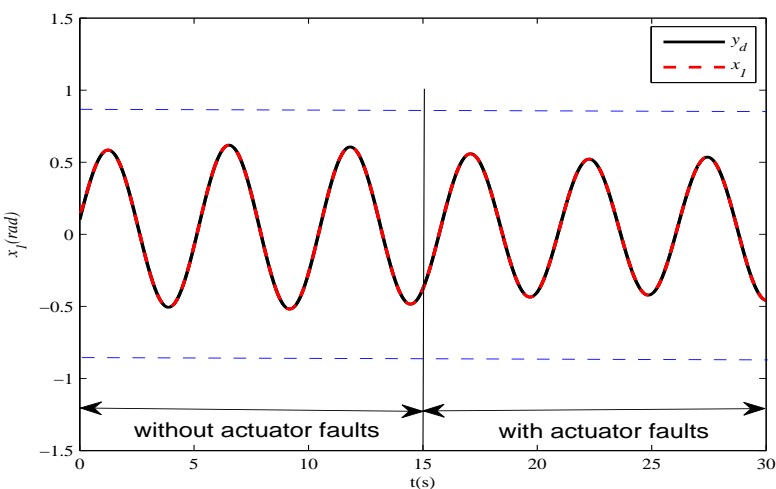

**Figure 4.** State $x_1$ tracking performance.

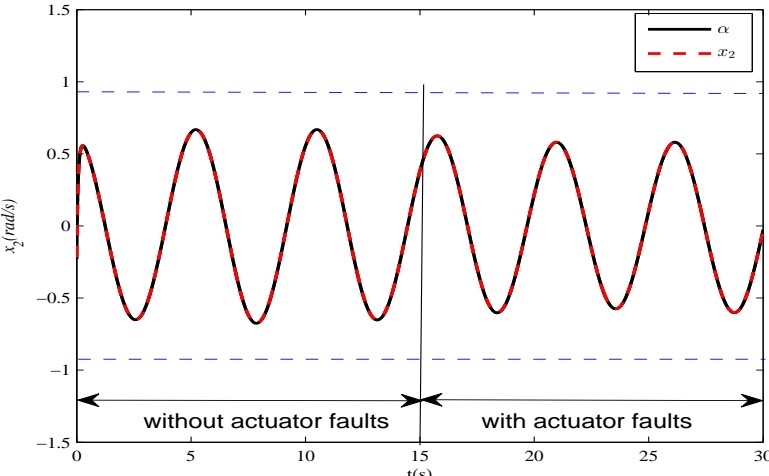

**Figure 5.** State $x_2$ tracking performance.

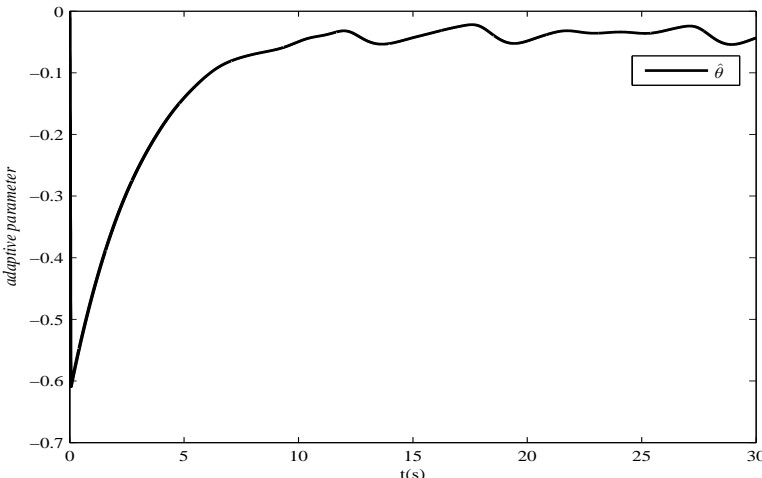

**Figure 6.** Adaptive parameter $\hat{\theta}$.

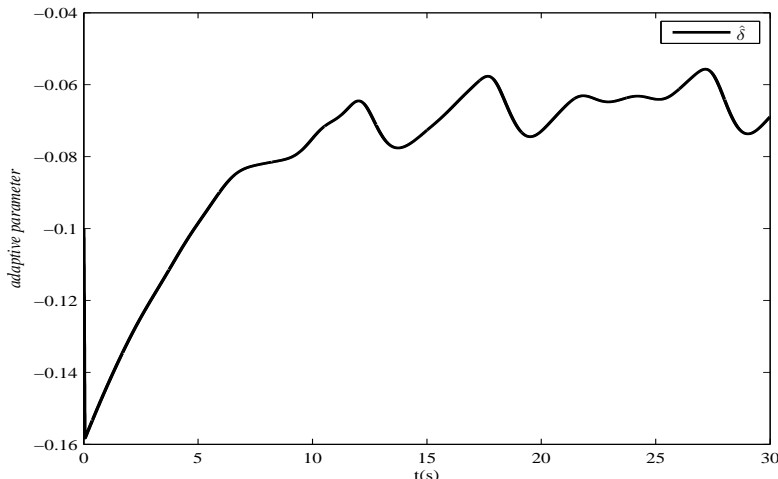

**Figure 7.** Adaptive parameter $\hat{\delta}$.

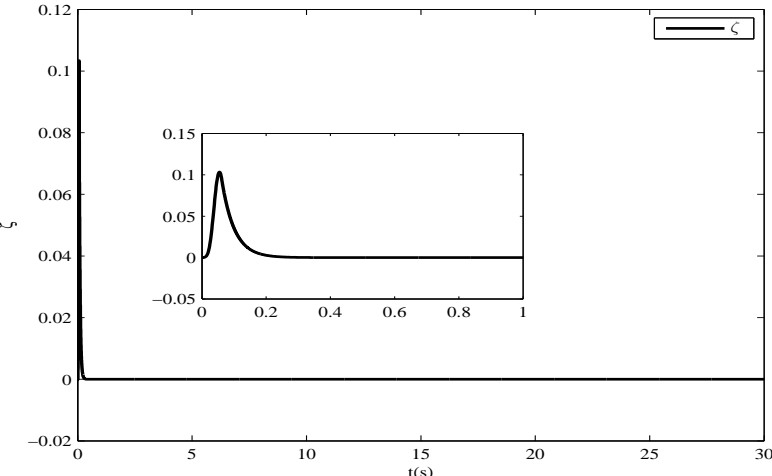

**Figure 8.** State of auxiliary system $\zeta$.

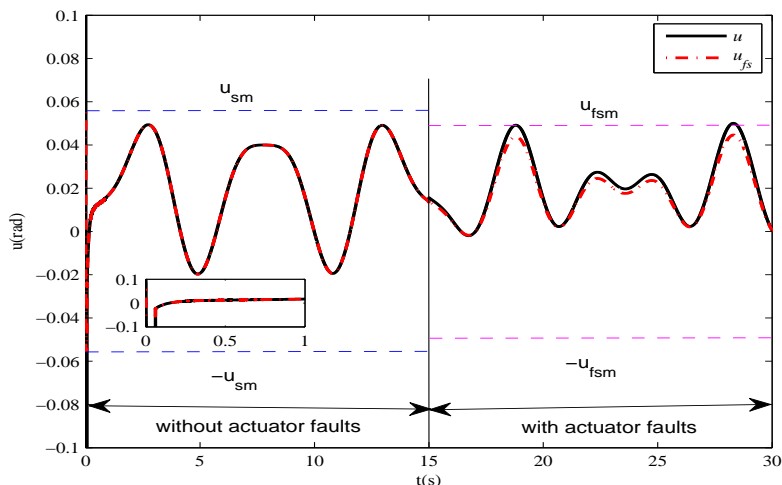

**Figure 9.** The control input with saturation and actuator faults.

## 3. Conclusions

An adaptive neural fault-tolerant control scheme was developed for a UAV helicopter SISO non-affine nonlinear yaw control system capable of dealing with unknown external disturbances, actuator faults, input saturation, and full-state constraints. By combining a Taylor series expansion technique with a robust sliding mode filter, the actual control input was explicitly defined. An RBFNN was employed to approximate the unknown nonlinear function. In comparison with prior research, this paper leads to the following salient conclusions:

(i) A symmetric barrier Lyapunov function with a smooth structure was designed to prevent the system from exceeding all state constraints.

(ii)To lighten the online computational burden, the Euclidean norm of the unknown neural network weight vector was estimated instead of the ideal weight vector. The number of learning parameters was reduced along with the complexity of the calculations.

(iii) The compound bounded disturbances and unknown parameters were estimated by adaptive technology. Moreover, we employed the limit of $g\rho$ to handle actuator fault tolerance, making it unnecessary to determine the bound value of $g\rho$ in the process of stability proofing. This broadens the range of applications.

(iv) By incorporating an anti-saturation compensator, the signal difference between actual control input and saturation actuator output was analyzed for its effect on system control so that the control input met the input constraint requirement.

The future investigative directions are to verify the feasibility of the UAV helicopter's longitudinal dynamics with existing time-varying state constraints, unknown control directions, and input saturation issues. From the perspective of the control system, we will design the overall solution of UAV helicopter based on the research in the paper and navigation instructions. In the future, the proposed control scheme should also extend its generality and adaptability to the MIMO systems.

**Author Contributions:** Conceptualization, Q.Z.; methodology, Q.Z. and X.C.; software, D.X.; validation, Q.Z. and X.C.; writing—original draft preparation, Q.Z., X.C. and D.X.; supervision, D.X. All authors have read and agreed to the published version of the manuscript.

**Funding:** This work was partially supported by National Natural Science Foundation of China (61503156, 61403161), Shandong Provincial Natural Science Foundation (ZR2019MF015) and the Key Research and Development Plan of Shandong Province of China (2017GGX30121).

**Conflicts of Interest:** The authors declare no conflict of interest.

## Appendix A. The Detailed Dynamic Equations of Helicopter Yaw-Channel Model

After complete employment with the assistance of Maple, we can obtain new expression of main rotor torque combined Equation (2) with $C_1 = \frac{1}{6}\rho abc\Omega^2(R^3 - R_0^3)$, $C_2 = \frac{1}{8}\rho abc\Omega\sqrt{2/\rho\pi R^2}(R^2 - R_0^2)$ ,where $R$ and $b$ are radial and number of the rotor.

$$
\begin{aligned}
Q_{mr} = {} & \frac{1}{8}C_{d2}\rho c\Omega^2(R^4 - R_0^4)\theta_{mr}^2 + \left[ 8C_{d2}\Omega\sqrt{\rho\pi R^2(2C_1\theta_{mr} + C_2^2 - C_2\sqrt{C_2^2 + 4C_2\theta_{mr}})}(R_0^3 - R^3) \right. \\
& + 4a\Omega\sqrt{\rho\pi R^2(2C_1\theta_{mr} + C_2^2 - C_2\sqrt{C_2^2 + 4C_1\theta_{mr}})}(R^3 - R_0^3) + 6C_{d2}C_1(R^2 - R_0^2) \\
& \left. + 6aC_1(R_0^2 - R^2) + 6C_{d1}\rho\pi\Omega^2 R^2(R^4 - R_0^4) \right]\frac{c\theta_{mr}}{48\pi R^2} + \left[ 3C_{d2}C_2\sqrt{C_2^2 + 4C_1\theta_{mr}}(R_0^2 - R^2) \right. \\
& + 3aC_2\sqrt{C_2^2 + 4C_1\theta_{mr}}(R^2 - R_0^2) + 4C_{d1}\Omega\sqrt{\rho\pi R^2(2C_1\theta_{mr} + C_2^2 - C_2\sqrt{C_2^2 + 4C_1\theta_{mr}})}(R_0^3 - R^3) \\
& \left. + 6C_{d0}\rho\pi\Omega^2 R^2(R^4 - R_0^4) + 3aC_2^2(R_0^2 - R^2) + 3C_{d2}C_2^2(R^2 - R_0^2) \right]\frac{c}{48\pi R^2}
\end{aligned}
$$

Likewise, the force which is created by the tail rotor can be expressed by the following form

$$
\begin{aligned}
T_{tr} &= \frac{1}{2}\rho a_{tr}b_{tr}c_{tr}\Omega_{tr}^2 \int_{R_{tr0}}^{R_{tr}} \left( \theta_{tr}r_{tr}^2 - \sqrt{\frac{T_{tr}}{2\rho A_{tr}}}\frac{r}{\Omega_{tr}} \right) dr_{tr} \\
&= C_3\theta_{tr} + \frac{1}{2}C_4(C_4 + \sqrt{C_4^2 + 4C_3\theta_{tr}})
\end{aligned}
$$

therein $C_3 = \frac{1}{6}\rho a_{tr}b_{tr}c_{tr}\Omega_{tr}^2(R_{tr}^3 - R_{tr0}^3)$, $C_4 = \frac{1}{8}\rho a_{tr}b_{tr}c_{tr}\Omega_{tr}\sqrt{2/\rho\pi R_{tr}^2}(R_{tr}^2 - R_{tr0}^2)$, where $a_{tr}, b_{tr}, c_{tr},$ $\Omega_{tr}, \theta_{tr}, r_{tr}$ are the slope of the lift curve, number of the rotor, chord of the blade, speed of the tail rotor, pitch angle, and radial distance, respectively.

In the same manner, the force produced by the main rotor can be formulated as

$$
T_{mr} = C_1\theta_{mr} + \frac{1}{2}C_2(C_2 + \sqrt{C_2^2 + 4C_1\theta_{mr}})
$$

We can see that the controller is hard to design so a model that can provide superior means of analysis and management is essential. By plotting the torque vs. pitch angle, we approximate the relation between $Q_{mr}$ and $\theta_{mr}$ with a quadratic polynomial [31].

$$
Q_{mr} = k_{Q_2}\theta_{mr}^2 + k_{Q_1}\theta_{mr} + k_{Q_0}
$$

Similarly, the force of the tail rotor $T_{tr}$ is described as

$$
T_{tr} = k_{T_2}\theta_{tr}^2 + k_{T_1}\theta_{tr} + k_{T_0}
$$

From the above analysis, we can get the non-affine nonlinear helicopter yaw-channel model as Equation (3).

## Appendix B. The Expression of Saturation Function $u_s$

In this paper, we mainly consider the typical saturated nonlinear function model, $u_s$ is described as follow

$$
u_s = sat(u) = \begin{cases} u_{sm}, & u \geq u_{sm} \\ u, & -u_{sm} < u < u_{sm} \\ -u_{sm}, & u \leq -u_{sm} \end{cases}
$$

where $u_{sm}$ is the maximum of saturation input $u_s$, $u$ denotes an actual control input to the system, $u_s$ and $u$ are the functions of time $t$.

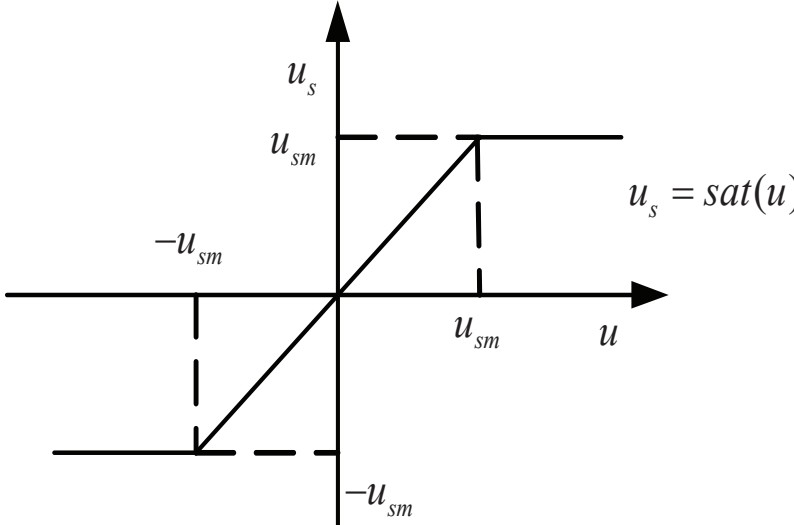

**Figure A1.** Saturation function.

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
