# Peer review of "Adaptive Neural Fault-Tolerant Control for the Yaw Control of UAV Helicopters with Input Saturation and Full-State Constraints"

_applsci, doi:10.3390/app10041404_

Round 1

Reviewer 1 Report

Dear Authors,

the paper deals with the design of an adaptive control system to control the yaw of an unmanned helicopter. Using the backstepping technique, authors design an RBF Neural Network based controller to achieve the stabilization of the system, considering uncertainties, disturbances, actuator faults, input saturations and full state constraints.

The paper is well written and explains definitions, lemmas and assumptions used in the design.

However, in my opinion, the paper is not complete considering the object to be controlled: the helicopter is a very complex system which dynamics on every axis is strongly coupled. While sometimes they are decoupled, using SISO controllers, the result is not optimal and must be taken into account.

Authors design a single SISO controller for yaw control, but the paper to be complete must consider the overall dynamics of the helicopter, in order to see the results of coupled dynamics. At this stage, although in remark 6 authors say that the same controller can be extended to MIMO systems, an idea can be to apply the same design also to pitch and roll, without considering coupling dynamics in the controller. It is important that the simulation results must be shown with maneuvers around every axes, in order to show any tracking error in 3D maneuvers.

For this reason, a major revision is needed.

Author Response

Dear reviewer,

      We greatly thanks for your positive assessments and constructive comments which have helped to improve our paper. In this round of major revisions, we have seriously considered each of your comments, and have revised the paper carefully according to your suggestions and comments. The main actions taken in response to your comments have been highlighted in red in the revised version. The details are explained as follows.

   1. For the non-affine nonlinear helicopter system, if the overall dynamic equations are taken into account, which are extremely complex. At the moment, the overall helicopter system has been decomposed into lateral and longitudinal dynamics in [R1]–[R3].
   2. This paper mainly studies cruising movement of the UAV helicopter system under the comprehensive factors of external disturbances, input saturation, all state constraints, and actuator fault. At present, little research has been done on such comprehensive issues. In this paper, the design of an adaptive neural network fault-tolerant controller not only ensures the stability of non-affine nonlinear UAV helicopter system in the process of lateral motion, but also guaran
tees the good tracking performance.
   3.  The lateral control scheme of UAV helicopter has been completed in this paper. In the established model, we take the pitch angle as control input of the system. By designing the control input of the UAV helicopter yaw-channel system reasonably, we can obtain that when input saturation and actuator faults occur simultaneously, the yaw-channel system can still track the desired           
trajectory.
    4.  Next, the future investigative direction is to verify the feasibility of the UAV helicopter’s longitudinal dynamics with existing time-varying state constraints, unknown control directions, and input saturation issues, and the proposed control scheme should be future extend its generality and adaptability to the multiple input multiple output system.

Reviewer 2 Report

I generally accept the method. However, I would prefer changing the presentation of the method through dividing the description into two parts: (1) Problem formulation, designing scheme, and computational examples; (2) Appendixes with theoretical considerations.

For instance, part (1) could consist of: (a) general models (1), (2), (8) with appropriate comments and references to Appendix 1; (b) models (9), (22), (23) with appropriate comments and references to Appendix 2; (c) the control scheme including steps 1, 2, 3 with most important formulas together with explanations; (d) results of computational experiments with exhaustive practically oriented comments.

Author Response

Dear reviewer,

  We greatly thanks for your careful assessments and constructive comments which have helped to improve our paper. We have revised the paper and focused our explanations on the points raised in your suggestions. The structure of this paper has been revised, the detailed revisions are marked in red in the paper. The revisions are divided into three parts:

1.In introduction, the frame of this paper has been revised. The paper contains three parts: introduction, main results and conclusions.

2.This paper mainly modified the main results section, which consists of four parts: UAV yaw-channel model, normal model, controller design and stability analysis, and simulation results.

3. The appendix A and appendix B have been added to the end of the paper, the detailed revisions are marked in red in the paper(see Appendix of the paper).

Round 2

Reviewer 1 Report

Dear Authors,

Even if the control technique is interesting, I don’t completely agree with the simplification of considering only the yaw channel, without taking into account pitch and roll.

1) Yes, the helicopter dynamics is usually decomposed into longitudinal and latero-directional dynamics, but the control system is considered always as unique. Coupling effects can be fundamental. They can be also neglected in the design of the control technique, but results must take into account them.

2) An overall control system must be designed and tested, otherwise the paper is incomplete. From an aeronautical point of view, the yaw channel is not the most important during cruise. I’d consider take-off or landing in presence of ground effect.

4) I appreciate the willingness to continue studying the field, but, again, it is very important to consider the overall system.

Furthermore, in the simulation results section, I suggest to explain the model used in the simulations or add more references. I cannot download “Zhao,X.;Han,J. Yaw control of helicopter: an adaptive guaranteed cost control approach. Inter Journ Innov Compu. Info. Cont. 2009, 5, 2267-2276.”. It is very important to test the technique with a full dynamic model and considering several maneuvers.

Author Response

Please see appendix.

Reviewer 2 Report

I accept the paper in its actual form, however, please correct the font in some parts of the text which unnecessarily are in italics.

Author Response

Dear reviewer,

    We greatly thanks for your positive assessment and constructive comment. we have seriously considered your comment, and have revised the paper carefully according to your suggestion. The main actions taken in response to your comment have been highlighted in the revised version. The details are portrayed as follows.

1、In main results, the fonts for parts of the normal model, derivation and description of the formulas in controller design and stability analysis, the proof of theorem 1, and all content in simulation results have been revised respectively.

2、The typefaces of the conclusions have been carefully checked and corrected.

3、We have updated the fonts of the author contributions, funding, conflicts of interest and references.

4、The font of the appendix has been examined and modified.

   Thank you again for your valuable comments.

Round 3

Reviewer 1 Report

Dear authors,

thank you for the complete answer to the comments.

After seen the updated version of the paper, it can be accepted.